# Asymbiotic Seed Germination and In Vitro Seedling Development of the Endangered Orchid Species *Cypripedium guttatum*

**DOI:** 10.3390/plants12223788

**Published:** 2023-11-07

**Authors:** Hyeong Bin Park, Jiae An, Kee-Hwa Bae, Seung Hyo Hong, Hwan Joon Park, Seongjun Kim, Chang Woo Lee, Byoung-Doo Lee, Ju Hyoung Baek, Nam Young Kim, Jung Eun Hwang

**Affiliations:** 1Research Center for Endangered Species, National Institute of Ecology, Yeongyang 36531, Republic of Korea; phb1274@nie.re.kr (H.B.P.); jiae_an@nie.re.kr (J.A.); sschow09@gmail.com (S.H.H.); rhg9281@nie.re.kr (H.J.P.); dao1229@nie.re.kr (S.K.); jacky903@nie.re.kr (C.W.L.); bdlee@nie.re.kr (B.-D.L.); iooju@nie.re.kr (J.H.B.); skadud2@nie.re.kr (N.Y.K.); 2Seed Vault Center, Baekdudaegan National Arboretum, Bonghwa 36209, Republic of Korea; khbae7724@koagi.or.kr

**Keywords:** *Cypripedium guttatum*, culture media, germination, light condition, organic supplementation, seeding growth

## Abstract

*Cypripedium guttatum* is a highly restricted terrestrial orchid that faces increasing endangerment owing to its habitat destruction and illegal collection. Compared to epiphytic orchids, terrestrial orchids such as *C. guttatum* have harder seed coats and more demanding in vitro germination conditions. This study aimed to develop an effective in vitro propagation system for *C. guttatum* to aid in its conservation. Seeds from mature capsules were subjected to various conditions, including sterilization using 1% sodium hypochlorite (NaOCl) and different light conditions, culture media, hormones, and organic supplements, to assess germination and early seedling development in vitro. Sterilization with 1% NaOCl significantly improved the germination rate, especially under dark conditions. Germination initiation occurred at 2 and 3 months in orchid seed sowing medium (OSM) and Murashige and Skoog (MS) medium, respectively. The addition of 1 mg/L naphthaleneacetic acid (NAA) further enhanced germination. However, the inclusion of organic supplements, such as apple and banana homogenates, in the culture medium led to substantial growth inhibition after 12 months. Notably, orchid maintenance medium (OMM) without organic additives proved to be the most suitable for seedling growth. The results of this study show that sterilization, appropriate light, and optimal NAA concentrations are beneficial for seed germination.

## 1. Introduction

With approximately 28,000 species, Orchidaceae is the largest family within the plant kingdom, constituting approximately 8% of angiosperm species diversity [1,2]. However, a significant number of orchid species are now endangered, primarily because of climate change, habitat loss, and their overexploitation for horticultural purposes [3,4]. The conservation and rejuvenation of the dwindling populations of these endangered orchids in their natural habitats are challenging because of the slow growth and low germination rates of orchids in the wild [5]. Generally, symbiotic and asymbiotic germination are used to propagate orchid seeds in vitro. In its habitat, orchid has a symbiotic relationship with mycorrhizal fungi for germination and seedling development. Thus, symbiotic germination requires the co-culturing of specific mycorrhizal fungi and orchid seeds for germination. Asymbiotic germination or in in vitro tissue culture allows for effective germination and propagation of orchid seeds by replacing mycorrhizal fungi with organic nutrients and culture medium. Therefore, in vitro tissue culture has emerged as a promising approach and is strongly recommended to prevent the extinction of these species.

*Cypripedium*, a temperate terrestrial orchid genus, commonly known as lady’s slipper due to its slipper-shaped flowers, comprises 56 species and four varieties [6,7]. However, the genus is becoming increasingly rare due to its habitat destruction and illegal collection [8]. Terrestrial orchid species, including *Cypripedium*, are generally more challenging to germinate in vitro than epiphytic species [9,10]. *Cypripedium*, *Paphiopedilum*, *Phragmipedium,* and *Selenipedium* are difficult to culture in vitro, with *Cypripedium* being the most recalcitrant [11]. Seeds of *Cypripedium* spp. generally have an impermeable, tough, two-layer seed coat, resulting in the inhibition of germination [12]. The layers of the seed testa have waterproof characteristics, and the integuments of the embryo contain germination inhibitors, such as abscisic acid (ABA) or suberin, that induce seed dormancy [13]. Thus, certain treatments are required to bypass water impermeability and physiological dormancy [14]. Zeng et al. [13] summarized the results of various studies on the in vitro seed germination of *Cypripedium* spp. Previous studies showed that seed germination and development in *Cypripedium* spp. are significantly influenced by various factors, such as capsule maturity, seed pretreatment, growth medium composition, and culture conditions. Several reports exist on the promotion of germination in *Cypripedium* by breaking the dormancy of seeds. Various strategies, such as prechilling [15], treatment with a NaOCl solution [16], and pretreatment with ultrasonication [17], have been employed to improve the seed germination rates and protocorm formation in *Cypripedium.* In particular, NaOCl effectively improved in vitro seed germination in *Cypripedium*. The pretreatment with NaOCl disintegrated the cell walls of the layers of the testa, which might have accelerated the absorbance of water by the embryos [13]. However, as each orchid sp. has different germination rates and in vitro seed germination characteristics, it is important to identify the optimal treatment conditions for specific orchid spp.

Although orchid seeds can germinate under both dark and light conditions, germination of terrestrial orchid seeds, including some of *Cypripedium*, was stimulated in the dark [9,18,19]. Generally, terrestrial orchids inhabit shaded areas in forests; therefore, after seed dispersal, the seeds may not receive enough light. Thus, as an adaptive mechanism, the germination of terrestrial orchids is considered to be advanced by darkness [14]. Zettler and Holfer [20] and Stewart and Kane [21] reported a higher germination of terrestrial orchids in complete darkness than in light. Also, the development of seedlings that germinated in darkness was higher than that of seedlings under light conditions. 

Treatments with plant growth regulators (PGRs) can induce seed germination in orchids. Exogenous cytokinin accelerated seed germination in terrestrial orchid spp. Some previous studies reported that cytokinin promoted seed germination and lipid mobilization in orchid embryos [14]. Seed germination in some members of *Cypripedium* was successfully stimulated by cytokinin. Auxins generally stimulate ethylene synthase and promote seed germination in many plants. However, the promotion of seed germination in terrestrial orchids by auxins has not been reported [14]. Miyoshi and Mii [15] reported that auxins did not increase seed germination but promoted protocorm development. However, to the best of our knowledge, there have been no studies on the influence of growth regulators on the germination rate and subsequent growth of *Cypripedium guttatum*. This provides an opportunity for further research in this field.

*C. guttatum* is a widely distributed slipper orchid that is distributed from easternmost Europe across Korea, Japan, and on to north-west America [22]. In general, *C. guttatum* primarily grows in high-altitude regions ranging from 1000 to 1400 m above sea level, but it can also be found at lower elevations in high-latitude areas such as Russia. It forms self-sustaining populations in open forests where the shrub or subshrub layers are underdeveloped. As a boreal plant species, *C. guttatum* is naturally occurring in North Korea, with numerous individuals, while within South Korea, it is currently restricted to the Hambaek mountain [23].

*C. guttatum* holds great significance in Korea’s flora. Classified as a Class 1 endangered wild species by the Ministry of Environment of the Republic of Korea, it is specially protected due to its indiscriminate harvesting for its horticultural value, which has led to a rapid decline in its habitat. Over recent decades, its habitats have declined, and this orchid is now found in a very limited number of sites on the Hambaek mountain (Figure 1a–d). The average temperature in its native habitat from July to August is about 19.3 °C. The altitude above sea level is 1271.2 m. The available soil depth is approximately 30 cm, and the soil was confirmed to be clay soil and loamy sand. The native habitat contains a vegetation composed of *C. macranthos*, *Lysimachia clethroides*, *Galium verum*, *Tripterygium regelii*, *Salix caprea*, *Quercus mongolica*, and *Fraxinus rhynchophylla* [23].

Therefore, establishing effective protocols for seed germination and subsequent seedling development is a prerequisite for habitat restoration and ex situ conservation. However, the approaches to promote seed germination and the establishment of *C. guttatum* seedlings are currently limited.

This study aimed to develop an efficient protocol for the asymbiotic seed germination of *C. guttatum*. This research focused on evaluating the effects of various culture variables, including medium type, light conditions, growth regulators, and organic additives, on seed germination and subsequent development. We hypothesized that (1) the germination rate would be increased by surface sterilization using NaOCl and dark conditions; (2) PGRs would affect the seed developmental stage, and the effect would be different according to the medium and hormone type used; (3) seedling growth would be different according to the medium and organic supplements provided.

## 2. Results and Discussion

### 2.1. Effects of Sodium Hypochlorite and Light and Dark Conditions on Seed Germination

The successful in vitro germination of terrestrial orchids necessitates the removal of inhibitory substances from the mature seeds, which is achieved by inducing embryo germination and protocorm formation through suitable surface sterilization methods. Previous research [24] indicated that this can be enhanced by treatment with NaOCl. In this study, we used two different media, half-strength Murashige and Skoog (1/2MS) medium and orchid seed sowing medium (OSM), to assess seed germination capacity. The germination rates were evaluated in these media for both the group treated with 1% NaOCl for 10 min and the untreated group. In the untreated group, no germination was observed in either medium. However, after treatment with 1% NaOCl, the germination rate exceeded 30% after 3 months in OSM and progressed at a low rate of 1% in 1/2MS (Table 1). Our findings demonstrated that sodium hypochlorite significantly promoted the formation of protocorms in *C. guttatum* (Table 1).

The enhancement of orchid seed germination might have been a result of the softening of the testa by the hypochlorite solution or cell wall-degrading enzymes, as well as by the leaching out of inhibitory substances with water [13]. The general, stimulatory effects of NaOCl solutions on surface disinfection can be attributed to several factors. One possible physiological effect is the removal of the endogenous inhibitory substance abscisic acid (ABA) from the seeds, disrupting dormancy and initiating germination. Another effect is the dissolution of substances on the embryonic surface, which facilitates water absorption [22]. Bae et al. [16] reported that the brown testa of *C. macranthos* turned white or transparent in the presence of 1.0% NaOCl for 30 min, and zygotic embryos inside the testas became visible. The authors demonstrated that only the whitened seeds could swell and germinate, whereas brown or unbleached seeds remained unchanged during cultivation. However, when sterilization was prolonged to the bleaching point, the seeds did not germinate. This was attributed to the higher concentration of the sterilizing substance or extended exposure time. Our results showed that a 10 min treatment with 1% NaOCl was the appropriate treatment to promote the germination of *C. guttatum*.

Generally, continuous darkness stimulates seed germination in terrestrial orchids, such as those of the *Habenaria* genus [22] and *Cypripedium* genus [25]. Fukai et al. [26] also indicated that darkness promotes the germination of mature seeds of the hybrid *Calanthe*. However, the effect of light on germination has not yet been investigated in *C. guttatum*. Therefore, in this study, we explored the germination rate under light and dark conditions. Germination was initially observed at 2 months under dark conditions and 1% NaOCl treatment in OSM and at 3 months in 1/2MS medium (Table 1). Under light treatment conditions, germination did not occur, regardless of the medium used and the NaOCl treatment. These results showed that, similar to other terrestrial orchid seeds [27,28], the germination of *C. guttatum* was inhibited by light exposure. Interestingly, in our study, germination did not occur when NaOCl or darkness was applied alone. These findings provided clear evidence that simultaneous NaOCl treatment and dark conditions promoted the germination of this orchid species. Additionally, in 1/2MS, the germination rate was significantly lower, at approximately 1%, even after 6 months (Table 1). To determine the differences between 1/2MS and OSM that caused the germination rate to be low in 1/2MS, the media compositions were compared, and growth regulators were applied.

### 2.2. Effects of Plant Growth Regulators on Seed Germination and Protocorm Development

Naphthaleneacetic acid (NAA) and 6-benzyladenopurine (BA) are well-known PGRs that are recognized for their potential effects on seed germination and early seedling development. This study extensively examined the specific effects of NAA and BA on germination rate, protocorm formation, and subsequent development of *C. guttatum*. The focus of this study was on understanding how BA and NAA influence seed germination, particularly in relation to protocorm development. Germination commenced 3 months after sowing in MS medium, yielding a germination rate of 1.02% (Table 2).

After 5 months, the germination rate significantly increased with the addition of NAA (33.28%) compared to that in MS (1.25%) and MS + BA (3.07%) (Table 2). Notably, the germination rates exhibited a substantial increase when BA and NAA were applied, with the most significant improvement observed in the presence of NAA. Conversely, in OSM, no significant difference was observed between the control medium and the media containing BA or NAA. This may be attributed to the fact that the basal OSM already contained 1 mg/L of NAA (Table 3). In 1/2MS + NAA medium and OSM containing the same amount of NAA, an increase in the germination rate of over 30% was observed at 5 and 3 months, respectively. This indicated that the germination rate varied depending on other medium components apart from the addition of NAA. The asymbiotic media used in this study contained similar components such as carbohydrates, inorganic salts, and gelling agents. However, there were variations in the specific mineral salts, organic additives, and vitamins in the media (Table 3). The mineral salts in the media differed not only in concentration, but also in available forms. Nitrogen is crucial for plant growth and development, and the choice of the nitrogen source was shown to influence the germination of various orchid species [21,29]. Anderson [29] suggested that organic nutrient sources might benefit the germination and development of *Platanthera ciliaris* because fungi naturally provide organic nitrogen to seeds. Stewart and Kane [21] demonstrated that different organic forms of nitrogen can influence the in vitro growth and development of *Habenaria* medusa. In this study, half-strength MS (1/2MS) medium contained only inorganic nitrogen sources (ammonium and nitrate), whereas OSM included a mixture of organic and inorganic nitrogen forms (Table 3). We found that germination and protocorm development of the studied orchids were promoted in OSM, suggesting that this species is highly capable of utilizing organic nitrogen sources.

Germination was observed across all treatment conditions three months after sowing, and a detailed analysis of protocorm development stages was conducted, documenting the morphological progression of *C. guttatum* from seed to early seedling (Figure 2a–g).

The percentage of seed germination was determined based on the testa rupture (Stage 2; Figure 2c). Immediately after the testa rupture (Stage 2; Figure 2c), a protocorm formed with a rhizoid (Stage 3; Figure 2d). Stage 4 protocorms showed the appearance of a protomeristem (Figure 2e), and by Stage 5, root elongation commenced (Figure 2f). At Stage 6, the development of shoots and roots progressed (Figure 2g). In the MS control medium (MS) and the BA-containing medium (MS + BA), over 90% of the seedlings were in developmental Stage 2. Conversely, in the medium containing NAA (MS + NAA), a higher number of seedlings were in Stage 3. Stage 5 development was not observed at 5 months in MS medium (Figure 3b,d). In contrast, in OSM, the developmental stages were evenly distributed at 3 months, and by 5 months, the majority of the seedlings had reached Stage 6 (Figure 3a,c).

Our results indicated that the germination and early seedling development of *Cypripedium guttatum* were influenced by the basal medium and the presence of NAA. Numerous studies showed that auxins, such as indole-3-acetic acid (IAA), NAA, and 2,4-dichlorophenoxyacetic acid (2,4-D), affect protocorm development in the orchid family. In certain orchid species, exogenous auxin application increased protocorm DNA content, diameter, morphology, and number during germination [30,31,32]. Miyosi and Mii [31] reported a significant increase in protocorm formation when mature seeds of *C. discolor* were presoaked in solutions containing NAA, ethephon (2-chloroethylphosphonic acid), or BA, compared to the control group. The observed enhancement ranged from 1.5 to 2.9 times, suggesting the substantial potential of these chemical treatments to significantly influence the germination and growth processes in *C. discolor* seeds [31]. Moreover, various orchid explants were reported to produce protocorm-like bodies (PLBs) in response to auxin and cytokinin, resembling protoplast formation during seed germination, albeit with varying optimal conditions for each species [33]. To the best of our knowledge, the promotive effect of NAA on seed germination has not been reported previously in members of the genus *C. guttatum*. Our study demonstrated that the application of 1 mg/L NAA promoted germination, protocorm formation, and development in this species. Future experiments should evaluate the effects of auxins other than NAA on seed germination in this species.

### 2.3. Effects of Organic Supplements on Seedling Growth

As demonstrated in previous studies, incorporating organic supplements into culture media can enhance the growth and development of various plant tissues in vitro. These supplements include apple juice, banana homogenates, coconut water, corn extracts, and potato homogenates [34,35]. These organic additives are integrated into the culture medium not only as a natural source of carbon, but also for their rich content in vitamins, fiber, hormones, and proteins [36]. In this study, we explored the effect of organic supplementation using an apple homogenate (A) and a banana homogenate (B) on seedling growth in *C. guttatum*. Surprisingly, our results indicated that the growth of *C. guttatum* seedlings was adversely affected by organic supplementation (Figure 4).

When the apple homogenate was added to the orchid maintenance medium (OMM), the fresh weight of the seedlings significantly decreased to about one-third compared to that in the control medium, and the root length also decreased by approximately two-fold (Figure 4c). In the case of banana homogenate supplementation, there was a slight reduction in both fresh weight and root length. Similarly, in MS basic medium, both apple and banana homogenates markedly suppressed the growth of *C. guttatum* (Figure 4). In the case of a simultaneous treatment with the apple and banana homogenates, most of the seedlings browned and died, and none progressed to Stage 6 of protocorm development (Figure 4a). These results suggested that organic supplements negatively affected the growth of *C. guttatum*. Contrary to these findings, organic supplements are known to improve orchid seed germination and seedling development in vitro. Similar observations were made for the development of *Paphiopedilum wardii* seedlings [37]. The authors observed that banana pulp had an inhibitory effect on seed development after the fifth stage (when two or more leaves were present). Additionally, Calevo et al. [38] reported the potential inhibitory effect of the organic supplement almond milk on the germination of *Cymbidium tracianum*. The inhibitory effect of organic supplements on early seedling growth may be attributed to the high sugar or nutrient content of such media, which hinders the effective growth of *C. guttatum*. To establish optimal growth conditions for *C. guttatum*, further research involving various concentrations of different supplements and hormones is imperative.

## 3. Materials and Methods

### 3.1. Plant Material and Sterilization

Mature seeds of *C. guttatum* were collected from the Hambaek mountain, Korea. The plants were hand-pollinated into different genotypes when they were in full bloom in mid-June (Figure 1a,b). The capsules were harvested from the plants 60 days after fertilization (16 August 2021; DAP 60; Figure 1c). After collection, the capsules were allowed to post-ripen at 25 ± 2 °C for one week and were subsequently stored in the dark at 4 °C until use. Mature capsules were thoroughly washed with Tween20 detergent under running tap water. Subsequently, they were soaked in 70% ethyl alcohol for 3 min and subjected to surface sterilization in a 12% sodium hypochlorite solution for 15 min. They were then rinsed five times for 1 min each with sterile water. The seed capsules were then carefully cut open, and the seeds were scraped out (26 August 2021).

### 3.2. Germination of the Seeds and Protocorm Development

Two germination media (OSM P723, PhytoTechnology, Lenexa, KS, USA) and 1/2MS (Duchefa, Haarlem, The Netherlands); Table 3 were used to evaluate the effects of surface sterilization, light conditions, and growth regulators on germination and protocorm development in *C. Guttatum*. The germination media were enriched with 20 g·L^−1^ sucrose, 1% activated charcoal, and 0.8% (w/v) agar. The pH was adjusted to 5.6 using 0.1N KOH before autoclaving at 121 °C for 15 min.

To investigate the influence of surface sterilization and light conditions on the germination rate, some seeds were treated with a 0% NaOCl solution, whereas others were exposed to a 1% NaOCl solution for 10 min and then rinsed five times with sterile water. After rinsing with sterile water, the seeds were sown using 1 mL of sterile water and a pipette. After sowing the sterilized seeds in the two basic media, the samples were divided and subjected to dark and light conditions. The protocorm percentage was calculated by counting the number of protocorms under a microscope every month. Five replicate plates were used for each treatment condition. To assess the effects of growth regulators on plant germination, 1 mg·L^−1^ of both BA and NAA was added to each basal medium. All the media were supplemented with 20 g·L^−1^ sucrose, 1% activated charcoal, and 0.8% (w/v) agar. The pH was adjusted to 5.6 using 0.1N KOH before autoclaving at 121 °C for 15 min.

The surface-sterilized seeds were evenly spread on the medium in each dish. Seed germination and protocorm development stages were graded from 0 to 6 (Table 4; modified from Kauth et al. [19]), and the rates were recorded monthly for 6 months. The first stage of germination consists of the swelling of the embryo, the subsequent rupture of the testa, and the development of the shoot and root. Germination is considered to have occurred when the embryo ruptures from the testa. The germination frequency of the seeds was scored using light microscopy.

The germination percentage was calculated by dividing the number of germinated seeds by the total number of seeds in the sample. The developmental stage percentage was calculated by dividing the number of seeds per germination stage by the total number of germinated seeds. At Stage 6, the plates (0/24 h L/D) were transferred from dark culture conditions to a 16/8 h L/D photoperiod.

### 3.3. In Vitro Seedling Development

This study investigated the impact of two basal media, OMM (PhytoTechnology, Lenexa, KS, USA; P668) and MS, along with the addition of organic supplements derived from apple and banana homogenates, on the in vitro seedling development of *C. guttatum*. Stage 6 protocorms were distributed per plate, with each treatment having ten replicates. To evaluate the effects of the organic supplements on seedling growth, the seedlings were cultivated in three separate media, which contained 30 g·L^−1^ of banana homogenate (B), 10 g·L^−1^ of apple homogenate, or a combination of the two homogenates. All media were enriched with 20 g·L^−1^ of sucrose, 1% activated charcoal, and 0.8% (w/v) agar before adjusting to pH of 5.6 using 0.1N KOH, followed by autoclaving at 121 °C for 15 min. The cultures were maintained at 25 ± 3 °C under a 16/8 h light/dark cycle. After a transfer period of 2 months, the fresh weight and length of the seedlings were measured. One-way analysis of variance (ANOVA) was used to assess differences in the germination rate, development stages, fresh weight, and root length. All results were considered statistically significant at *p* ≤ 0.05. All statistical analyses were performed using SAS 9.4 statistical software (SAS Institute Inc., Cary, NC, USA).

## 4. Conclusions

Our study provides compelling evidence that a substantial number of *C. guttatum* seedlings can be generated in vitro using asymbiotic germination techniques. In vitro germination and the seedling developmental stages were significantly influenced by photoperiod, surface sterilization, and the choice of asymbiotic medium. We observed a significant enhancement in the germination of *C. guttatum* when the seeds were surface-sterilized with 1% sodium hypochlorite and sown on a culture medium supplemented with NAA, followed by placement in dark conditions. The most suitable media for the germination and growth of this species were OSM and OMM. Seed germination and protocorm development were promoted by the OSM containing both organic and inorganic forms of nitrogen sources. Interestingly, our results indicated that organic supplements hindered the growth of this species. However, to optimize the growth conditions for *C. guttatum,* further research is warranted to identify beneficial supplements and their appropriate concentrations. In conclusion, this study presents pivotal findings to effectively support the germination and seedling development of *C. guttatum.* The insights provided here are invaluable to researchers engaged in the conservation and restoration efforts related to this plant.

## Figures and Tables

**Figure 1 plants-12-03788-f001:**
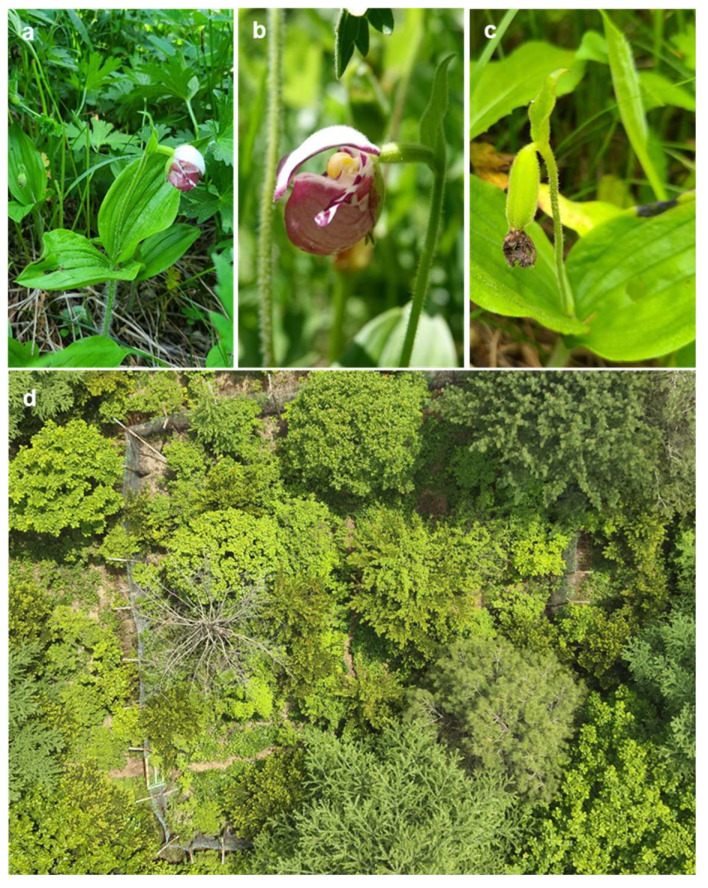
(**a**) *Cypripedium guttatum* plants, (**b**) flower, (**c**) maturing capsules, (**d**) typical habitat of *Cypripedium guttatum* on Hambaek mountain.

**Figure 2 plants-12-03788-f002:**
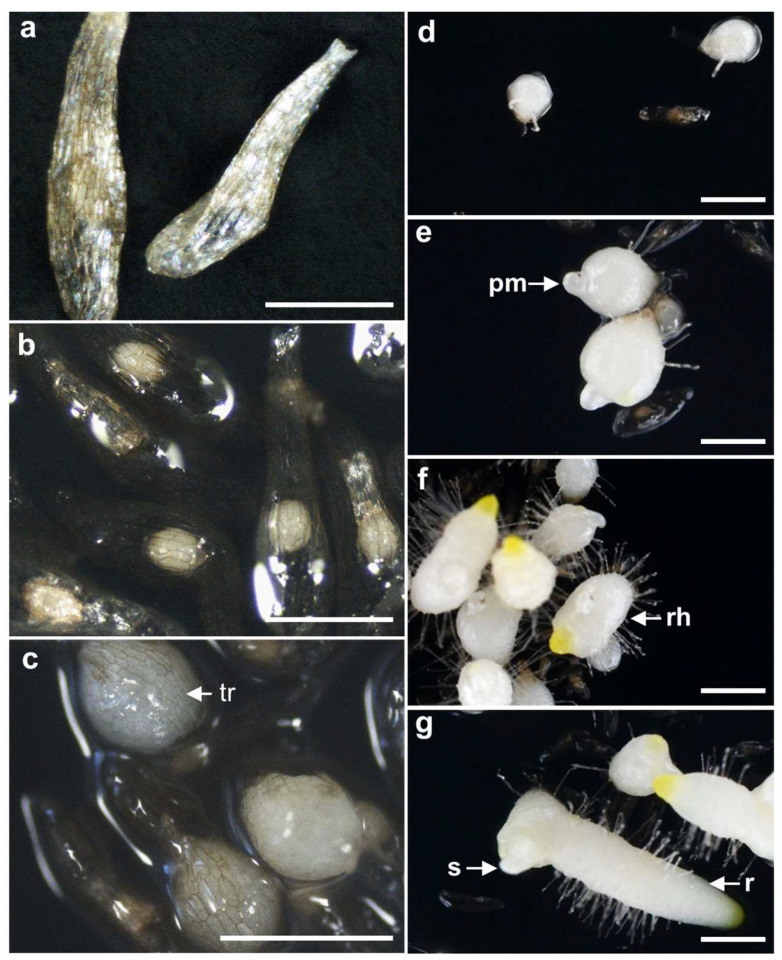
In vitro asymbiotic germination of *Cypripedium guttatum*. (**a**) Stage 0 “no germination”; (**b**) Stage 1, “pregermination”; (**c**) Stage 2, “germination” (tr = testa ruptured); (**d**) Stage 3, “protocorms”; (**e**) Stage 4 “appearance of protomeristem” (pm = protomeristem); (**f**) Stage 5 “pointed shoot apex and rhizoids” (rh = rhizoids); (**g**) Stage 6 “elongation of root” (s = shoot, r = root). Bars: 0.5 mm (**a**–**c**); 1 mm (**d**–**g**).

**Figure 3 plants-12-03788-f003:**
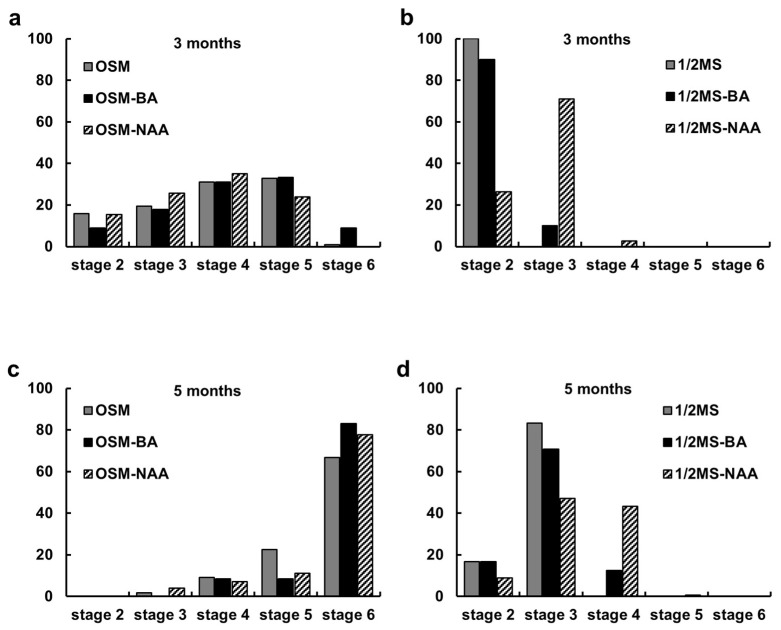
Comparative effects of culture media and growth regulators on the in vitro seedling development stage of *Cypripedium guttatum* after 3 (**a**,**b**) and 5 (**c**,**d**) months of asymbiotic culture. OSM: orchid seed sowing medium (PhytoTechnology, P723); 1/2MS: half-strength Murashige and Skoog medium; BA: 6-benzyladenopurine; NAA: α-naphthalene acetic acid.

**Figure 4 plants-12-03788-f004:**
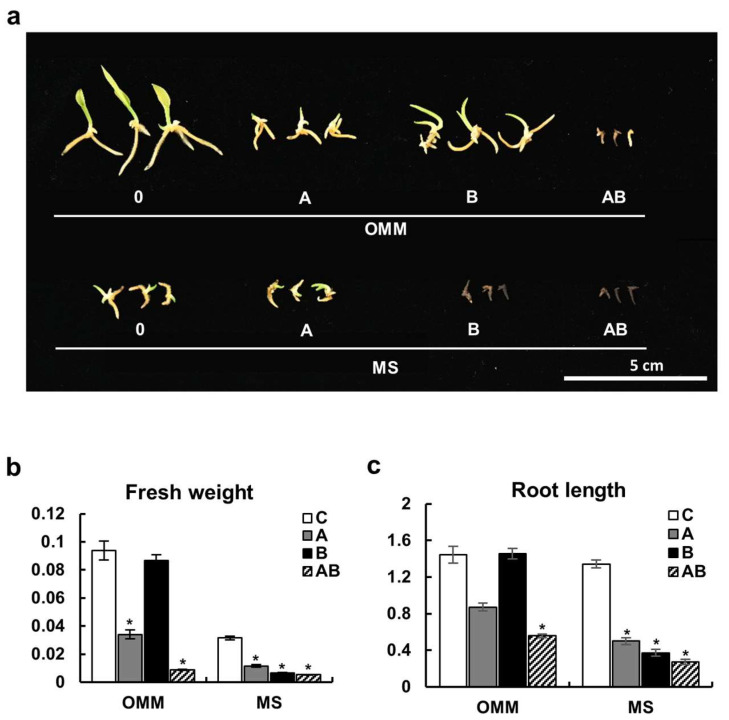
Effect of apple and banana homogenates on *Cypripedium guttatum* seedling growth. (**a**) Growth differences in seedlings were determined seven months after sowing; (**b**) Fresh weight and (**c**) root length were measured. Scale bar = 5 cm. MS: full-strength Murashige and Skoog medium; OMM: orchid maintenance medium; A: apple homogenate; B: banana homogenate. The data are presented as the mean ± SD of the values obtained from triplicate experiments. * *p* ≤ 0.05.

**Table 1 plants-12-03788-t001:** Germination rates of *Cypripedium guttatum* seeds according to culture medium, NaOCl concentration, and light conditions.

Treatment	Germination (%)											
	1/2MS					OSM						
	1 (M)	2 (M)	3 (M)	4 (M)	5 (M)	6 (M)	1 (M)	2 (M)	3 (M)	4 (M)	5 (M)	6 (M)
NaOCl (0%) + Light	-	-	-	-	-	-	-	-	-	-	-	-
NaOCl (1%) + Light	-	-	-	-	-	-	-	-	-	-	-	-
NaOCl (0%) + Dark	-	-	-	-	-	-	-	-	-	-	-	-
NaOCl (1%) + Dark	-	-	1.02	1.25	1.25	1.35	-	9.76	31.52	33.43	33.43	33.43

M: month(s), 1/2MS: half-strength Murashige and Skoog medium, OSM: orchid seed sowing medium.

**Table 2 plants-12-03788-t002:** Germination rates of *Cypripedium guttatum* seeds grown in different culture media according to the addition of naphthaleneacetic acid (NAA) and 6-benzyladenopurine (BA).

Medium	Germination (%)					
	1 (M)	2 (M)	3 (M)	4 (M)	5 (M)	6 (M)
1/2MS	-	-	1.02	1.25	1.25	1.36
1/2MS + BA	-	-	4.03	3.96	6.07	6.07
1/2MS + NAA	-	-	9.34	23.34 *	33.28 *	34.27 *
OSM	-	9.77	31.52	33.43	33.43	33.43
OSM + BA	-	11.75	35.75	37.41	37.41	37.41
OSM + NAA	-	4.55	33.02	36.13	36.13	36.13

M: month(s), 1/2MS: Murashige and Skoog medium, OSM: orchid seed sowing medium. The asterisks (*) indicate significant differences (* *p* ≤ 0.05).

**Table 3 plants-12-03788-t003:** Nutrient composition of the germination media used for the asymbiotic seed germination of *Cypripedium guttatum*.

	OSM	1/2MS
Macronutrients (mM)		
Ammonium Nitrate	5.15	10.31
Calcium	0.75	1.50
Chlorine	1.50	3.1
Magnesium	0.62	0.75
Nitrogen–Nitrate	9.85	19.70
Potassium	5.62	10.89
Phosphate	0.31	0.63
Sulfate	0.71	0.86
Sodium	0.10	0.10
Micronutrients (μM)		
Boron	26.7	50
Cobalt	0.026	0.053
Copper	0.025	0.5
Iron	50	50
Iodine	1.25	2.50
Manganese	25	50
Molybdenum	0.26	0.52
Zinc	9.22	14.95
Organics (mg/L)		
*myo*-Inositol	100	
Nicotinic acid	1.0	
Peptone	2000	
Pyridoxine · HCl	1.0	
Thiamine · HCl	10	
Total mineral salt concentration (mM)	24.72	48.01
Total inorganic N (mM)	15.00	30.01
NH_4_:NO_3_	0.52	0.52

OSM; P723 PhytoTechnology orchid seed sowing medium, 1/2MS; half-strength Murashige and Skoog, (modified from [14]).

**Table 4 plants-12-03788-t004:** Seed germination and protocorm developmental stages of *Cypripedium guttatum* (modified from Kauth et al. [19]).

Stage	Description
0	No germination
1	Embryo swollen
2	Embryo ruptured through testa (=germination)
3	Enlarged seed without testa
4	Appearance of protomeristem
5	Elongation of root
6	Development of shoot and root

## Data Availability

Data are contained within the article.

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
