# Peer review of "Asymbiotic Seed Germination and In Vitro Seedling Development of the Endangered Orchid Species *Cypripedium guttatum"

_plants, 2023, doi:10.3390/plants12223788_

Round 1
Reviewer 1 Report
Comments and Suggestions for Authors
The manuscript ID: plants-2684067 entitled "Asymbiotic seed germination and in vitro seedling development of the endangered orchid species Cypripedium guttatum” submitted to Plants has been reviewed. This study provides valuable insights for the asymbiotic seed germination of Cypripedium guttatum. The authors report that this species is under special protection in Korea due to the decreasing number and area of habitats as a result of habitat destruction and illegal plant collection. The presented research was aimed at developing an effective protocol for asymbiotic seed germination and early development of C. guttatum seedlings. The influence of the type of substrate, light conditions, growth regulators and organic additives on seed germination and the subsequent development of seedlings of this species in vitro was examined. It has been shown that orchid care medium (OMM) without organic additives was the most suitable for the growth of seedlings. Moreover, it was found that sterilization, adequate light and optimal NAA concentration are crucial for the germination of seeds of the tested species. The obtained results of this research contribute to a better scientific understanding of seed germination and seedling development of C. guttatum and are a prerequisite for habitat restoration and ex situ protection of this orchid species.
In general, I found this paper to be well-written, with interesting results and conclusions, but there are a few points that need attention, see below.
1. The authors did not formulate any hypotheses only aims.
2. The Material and Methods section should be placed first, followed by the Results and Discussion section.
3. Personally, I prefer when the Discussion is a separate chapter, rather than combined with the results.
4. Line 60: Cypripedium should be in italics
5. Line 87 and 88: It is not known whether the species Cypripedium guttatum or its variety was tested? Line 87 states that it is one of the smallest species, while line 88 states that C. guttatum variety holds great significance in Korea's flora. Please explain this.
6. The introduction should include brief information about the habitat conditions of the orchid under study: soil conditions, pH, nutrient content (macroelements), habitat humidity, accompanying species. Figure 1(d) doesn't show much.
7. The Materials and Methods chapter does not specify when the research was conducted
8. What weather conditions prevailed during the seed setting and ripening period, these conditions affect the seed germination capacity
9. The results only show germination (in %), but what about the remaining seeds, were they dead or freshly ungerminated?
10. The Materials and methods chapter also lacks information about the statistical analysis of the research results.

Minor editing of English language required
Reviewer 2 Report
Comments and Suggestions for Authors
There are some suggestions.
1. Since "asymbiotic" is specifically used for seed germination of C. guttatum, please add some information on the difference between symbiotic and asymbiotic germination in the introduction.
2. In line 55-56, the word "in" may be redundant in the sentence "Several reports exist on the promotion of Cypripedium by breaking the dormancy of seeds."
3. In line 60-61, the word "and" may be redundant in the sentence "The pretreatment with NaOCl disintegrated the cell walls of the layers of the testa, which might have accelerated the absorbance of water by the embryos."
4. In line 79-80, correct the sentence as, "However, the promotion of seed germination in terrestrial orchids by auxins has not been reported."
5. In line 134-135, “Our results affirmed that a 10 min treatment with 1% NaOCl was the appropriate concentration to promote the germination of C. guttatum”. I suggest that “concentration” can be replaced by “combination or treatment” to express the treatment of “a 10 min treatment with 1% NaOCl”.
6. In line 244, “using banana homogenate (B) and apple homogenate (A) on” is suggested to be revised as “using apple homogenate (A) and banana homogenate (B) on”.
7. In line 343-345, the authors stated that "The most suitable media for the germination and growth of this species were OSM and OMM containing nitrogen sources in both organic and inorganic forms." But germination was not tested on OMM medium, and the experiment regarding "OSM and OMM containing nitrogen sources in both organic and inorganic forms" was not conducted in this work. Please revise this statement.
Reviewer 3 Report
Comments and Suggestions for Authors
This article describes a series of experiments conducted by the authors to explore factors important in improving seed germination and seedling development in an endangered orchid species. Although I am not a specialist in orchids, I have published some papers on seed germination. I found the orchid aspect to be interesting and important.
As the only important comment, I suggest you mention somewhere about the number of seeds tested. It seems like they were very small, and each of your 5 replicates in each treatment were probably different numbers, but presumably you had some type of range to keep the numbers somewhat similar? So 50 per replicate? 500? 5000? Just a sentence giving a little more information would be beneficial.
Otherwise, I only have minor comments, as the paper was generally well-written and easy to understand
L28; suggest "crucial" --> beneficial
L34; delete "the"
L54: by various factors
L55; of germination in
L59; effectively caused sterilization -- I'm not sure what you mean by this. Do you mean surface sterilization? If so, how did you quantify its effectiveness?
L62; delete "and"
L63;represented --> have
L66/67: produced seeds, when germination was stimulated in the dark... I am not sure what this means? Do you mean the parent plant produced seeds?
L70; higher germination in the dark / light inhibits germination --> this sentence could be more concise and shorter. It feels repetitive.
L87; a significant morphological position. -- I don't know what that means.
L89; Specify which country's Ministry of Environment (presumably S Korea?)
L121; can be facilitated by --> may be a result of
L145 (and probably elsewhere) is --> was
L179; delete "nutrition"
L254; more than 3-fold -- I suggest you say "to about 1/3" or similar.
Comments on the Quality of English LanguageThe language was broadly of a good quality anf the paper was easy to understand. Congratulations!
